# Singular Cauchy Problem for a Nonlinear Fractional Differential Equation

Victor Orlov 

Institute of Digital Technologies and Modeling in Construction, Moscow State University of Civil Engineering, Yaroslavskoye Shosse, 26, 129337 Moscow, Russia; orlovvn@mgsu.ru; Tel.: +7-916-457-4176

**Abstract:** The paper studies a nonlinear equation including both fractional and ordinary derivatives. The singular Cauchy problem is considered. The theorem of the existence of uniqueness of a solution in the neighborhood of a fixed singular point of algebraic type is proved. An analytical approximate solution is built, an a priori estimate is obtained. A formula for calculating the area where the proven theorem works is obtained. The theoretical results are confirmed by a numerical experiment in both digital and graphical versions. The technology of optimizing an a priori error using an a posteriori error is demonstrated.

**Keywords:** nonlinear differential equation; fractional derivative; singular points; analytical approximate solution; a priori estimate

**MSC:** 34G20; 35A05

## 1. Introduction

A large number of problems in various fields lead to mathematical models based on differential equations. Thus, we encounter linear ordinary differential equations when solving the problem of body motion around a fixed point (Euler's dynamic equation) [1], in describing the dependence of a body's acceleration on the resultant of all applied forces (Newton's equations on a straight line), and when finding a curve of uniform descent (Jacob Bernoulli equation) [2]. Problems in fields such as control and optimization theory [3], quantum field theory [4], as well as the simplest liquid nonlinear oscillations, lead to the nonlinear differential equations of Ricatti. Abel's equations arise in problems of nonlinear optics when describing a superradiant avalanche [5], the theory of finite elasticity [6], nonlinear diffusion [7], and the nonlinear thermal conductivity of a steady state [8].

Moving algebraic-type singular points play an important role in nonlinear differential equations. Such equations belong to the class of equations that are generally not solvable in quadratures. The solution region is approached in two domains, the neighborhood of singular points and the analyticity area.

At the moment, based on publications, two approaches to solving nonlinear differential equations can be noted.

The first approach involves special substitutions of variables that, with certain parameter values, allow for the resolution of nonlinear differential equations in quadratures. This approach allows us to resolve nonlinear differential equations only in special cases [9–19].

The second approach is related to the search for an analytical approximate solution presented in the author's works. Theorems of existence and uniqueness of solutions are proved both in the area of analyticity and in the neighborhood of moving singular points. Analytical approximate solutions are constructed in these specified regions, with techniques presented for optimizing a priori estimates through the use of a posteriori estimates [20–25]. The proposed idea has been successfully implemented for other classes of nonlinear differential equations [26–28].

Up to now, only linear equations have been considered for equations with fractional derivatives and the features of methods for solving explicit and implicit schemes combining types of Riemann–Liouville and Caputo derivatives have been investigated.

At the moment, there are no works devoted to the study of nonlinear differential equations with fractional derivative. It is necessary to note the features of the theory for ordinary differential equations (ODE) and equations with a fractional derivative (EFD). The theorem of the existence and uniqueness of the solution to the ODE has a local character, and for the EFD—a global character.

In this work, the idea of the author's analytical approximate solution method [25] for a nonlinear equation with a fractional derivative is implemented.

Taking into account Fuchs' classification of singular points into fixed and moving [29], the problem of solving a nonlinear differential equation divides the search area for a solution into two parts. The first part concerns the analyticity region of the solution, and the second part concerns the neighborhood of singular points.

It should be noted that the classical theory of existence and uniqueness of solutions to linear equations with fractional derivatives does not work in this case.

The article contains a review of sources on the topic under consideration with an analysis of research solution options. The research methods section includes the implementation of the author's approach to finding an analytical approximate solution in the neighborhood of a fixed singular point [25] for a nonlinear differential equation with fractional derivatives for the case of an equation with fractional derivatives. The theorem of the existence and uniqueness of a solution in the neighborhood of a fixed singular point of the equation under consideration is proved. At the first stage of the proof, the uniqueness of the solution structure is substantiated, and at the second stage, the convergence of the correct part of the constructed series is proved. An analytical approximate solution in the neighborhood of a fixed singular point is built. A formula for calculating the region in which the theorem works is obtained. A numerical experiment is presented in the results discussion section. The calculation results are presented both numerically and graphically. The calculations were performed in Excel. The conclusion lists the obtained results and prospective problems for subsequent research.

## 2. Research Method: Statement of the Singular Cauchy Problem

The movement of fluid in fractured reservoirs is described by the following formula [30]:

$$\nabla P = \frac{\mu}{k} \cdot \overline{v} + \frac{\mu \beta v}{k} \cdot \overline{v}, \tag{1}$$

here, $\nabla P$—is the pressure gradient; $\mu$—dynamic viscosity of the liquid; $v$—filtration rate, $k$—permeability of the medium; $\beta$—coefficient depending on the geometry of the fractured medium and on the micro roughness, which is determined empirically [31].

A continuity equation for an incompressible fluid, with radial filtration to a vertical cylindrical well, can be written as follows:

$$\mathrm{div}\,\overline{v} = 0. \tag{2}$$

Equation (2) in a cylindrical coordinate system has the following form:

$$\frac{1}{r} \cdot \frac{\partial(rv_r)}{\partial r} + \frac{1}{r}\frac{\partial(rv_\theta)}{\partial \theta} + \frac{\partial(v_z)}{\partial z} = 0, \tag{3}$$

where $r \geq 0$.

For an axisymmetric and planar problem, Equation (3) takes the following form:

$$\frac{1}{r} \cdot \frac{\partial(rv_r)}{\partial r} = 0. \tag{4}$$

Expressing $v$ from Equation (1), we obtain

$$v = \frac{\sqrt{1 + 4\beta k |\nabla P| / \mu} - 1}{2\beta}.$$ (5)

Substituting Equation (5) into Equation (4), we obtain

$$\frac{d}{dr}\left[r\left(\sqrt{1 + \frac{4\beta k |\nabla P|}{\mu}} - 1\right)\right] = 0.$$

Obviously, there exists a constant $C_1$ such that

$$r\left(\sqrt{1 + \frac{4\beta k |\nabla P|}{\mu}} - 1\right) = C_1.$$

We find the integration constant $C_1$ under the condition $|\nabla P| < |\nabla P|_{crit}$ and, knowing the flow rate of a well with a radius of $r_c$, we open a layer with a thickness of $H$:

$$Q = 2\pi (rHv)_{r=r_c}.$$ (6)

Substituting expression (5) into (6), we obtain $Q$, the value of the volume of liquid over a certain period of time. The flow goes in one direction:

$$Q = \frac{\pi r H}{\beta}\left(\sqrt{1 + \frac{4\beta k}{\mu}|\nabla P|} - 1\right).$$ (7)

In the work [32] a plane axisymmetric problem of steady fluid flow in the vicinity of a well was considered, revealing a purely fractured dependence of the effective thickness of the formation on the pressure gradient and was approximated by the following dependence:

$$H = hD^\alpha \frac{P}{(|\nabla P|_{crit})^\alpha}, \quad 0 < \alpha < 1.$$ (8)

where $H$ is the thickness of the formation, $h$—is the thickness of the formation at $|\nabla P|_{crit}$; $|\nabla P|_{crit}$ is a pressure gradient; $\alpha$—is an empirical coefficient characterizing the change in the effective thickness of the formation from the pressure gradient.

If we substitute expression (8) into relation (7), we obtain

$$\sqrt{1 + \frac{4\beta k}{\mu}|\nabla P|} - 1 = \frac{Q\beta |\nabla P|_{crit}^\alpha}{\pi r h D^\alpha P}.$$ (9)

Having performed a number of transformations in Equation (9), we arrive at the following form:

$$[D^\alpha P]^2 |\nabla P| = a(r)D^\alpha P + b(r), \quad 0 \le \alpha < 1,$$

where

$$a(r) = \frac{\mu Q |\nabla P|_{crit}^\alpha}{2\pi r h k}; \quad b(r) = \frac{\mu \beta}{k}\left(\frac{Q |\nabla P|_{crit}^\alpha}{2\pi r h}\right)^2.$$

$r \in [r_c, r_k]$, $r_c$ is the well radius, $r_k$ is the supply contour radius (formation boundary). Let us introduce the following notations:

$$a(r) = A_1 \frac{1}{r}, \quad b(r) = B_1 \frac{1}{r^2},$$

where $A_1$ and $B_1$ are constants, respectively, are equal to the expressions:

$$A_1 = \frac{\mu Q |\nabla P|_{crit}^{\alpha}}{2\pi hk}, \quad B_1 = \frac{\mu \beta}{k} \left( \frac{Q |\nabla P|_{crit}^{\alpha}}{2\pi h} \right)^2.$$

Let us consider the singular Cauchy problem:

$$|P'(r)| [D^{\alpha} P(r)]^2 = A_1 \frac{1}{r} D^{\alpha} P(r) + B_1 \frac{1}{r^2}. \tag{10}$$

$$P(0) = \infty. \tag{11}$$

We proceed to search for a solution to problem (10), (11) in the way

$$P(r) = r^{\rho} \sum_{n=0}^{\infty} m_n r^{n/10} = m_0 r^{\rho_1} + r^{\rho_2} \sum_{n=1}^{\infty} m_n r^{n/10}, \tag{12}$$

where $n = 0, 1, 2, 3\ldots, \rho_1, \rho_2$, are the parameters to be determined.

Let us determine $P'(r)$:

$$P'(r) = m_0 \rho_1 r^{\rho_1 - 1} + \sum_{n=1}^{\infty} m_n (n/10 + \rho_2) r^{n/10 + \rho_2 - 1} = \tag{13}$$

$$m_0 \rho_1 r^{\rho_1 - 1} + \sum_{n=1}^{8} m_n (n/10 + \rho_2) r^{n/10 + \rho_2 - 1} + \sum_{n=9}^{\infty} m_n (n/10 + \rho_2) r^{n/10 + \rho_2 - 1}.$$

For a fractional derivative we will have the following:

$$D^{\alpha} P(r) = m_0 D^{\alpha} r^{\rho_1} + \sum_{n=1}^{\infty} m_n D^{\alpha} r^{n/10 + \rho_2} =$$

$$m_0 \frac{\Gamma(\rho_1 + 1)}{\Gamma(\rho_1 + 1 - \alpha)} r^{\rho_1 - \alpha} + \sum_{n=1}^{\infty} m_n \frac{\Gamma(n/10 + \rho_2 + 1)}{\Gamma(n/10 + \rho_2 + 1 - \alpha)} r^{n/10 + \rho_2 - \alpha} = \tag{14}$$

$$m_0 g_0 r^{\rho_1 - \alpha} + \sum_{n=1}^{\infty} m_n g_n r^{n/10 + \rho_2 - \alpha},$$

where

$$\frac{\Gamma(\rho_1 + 1)}{\Gamma(\rho_1 + 1 - \alpha)} = g_0, \quad \frac{\Gamma(n/10 + \rho_2 + 1)}{\Gamma(n/10 + \rho_2 + 1 - \alpha)} = g_n.$$

Let us define

$$C_n = m_n g_n. \tag{15}$$

Then, from (14) for $[D^{\alpha} P(r)]^2$ we obtain

$$[D^{\alpha} P(r)]^2 = C_0^2 r^{2(\rho_1 - \alpha)} + 2C_0 \sum_{n=1}^{\infty} C_n r^{n/10 + \rho_1 + \rho_2 - 2\alpha} + \sum_{n=1}^{\infty} C_n^* r^{n/10 + \rho_2 - \alpha} \tag{16}$$

where

$$C_n^* = \sum_{i=1}^{n} C_i C_{n-i}.$$

Taking into account (13) and (16), the left side of Equation (10) will have the following form:

$$|P'(r)|[D^\alpha P(r)]^2 = C_0^2 m_0 \rho_1 r^{3\rho_1 - 2\alpha - 1} - C_0^2 \sum_{n=1}^{\infty} m_n(\frac{n}{10} + \rho_2) r^{n/10 + 2\rho_1 + \rho_2 - 2\alpha - 1} +$$

$$2m_0 C_0 \rho_1 \sum_{n=1}^{\infty} C_n r^{n/10 + 2\rho_1 + \rho_2 - 2\alpha - 1} - 2C_0 r^{2\rho_2 + \rho_1 - 2\alpha - 1} \sum_{n=1}^{\infty} C_n r^{n/10} \sum_{n=1}^{\infty} m_n(\frac{n}{10} + \rho_2) r^{n/10} + \tag{17}$$

$$m_0 \rho_1 r^{2\rho_2 + \rho_1 - 2\alpha - 1} \sum_{n=1}^{\infty} C_n^* r^{n/10} - r^{3\rho_2 - 2\alpha - 1} \sum_{n=1}^{\infty} C_n^* r^{n/10} \sum_{n=1}^{\infty} m_n(\frac{n}{10} + \rho_2) r^{n/10}.$$

Let us expand the right side of Equation (10):

$$A_1 \frac{1}{r} D^\alpha P(r) + B_1 \frac{1}{r^2} = A_1 \sum_{n=0}^{\infty} C_n r^{n/10 - \rho_1 - \alpha - 1} + B_1 r^{-2}. \tag{18}$$

Taking into account (17) and (18), Equation (10) in its expanded form has the following form:

$$-C_0^2 m_0 \rho_1 r^{3\rho_1 - 2\alpha - 1} - C_0^2 \sum_{n=1}^{\infty} m_n(\frac{n}{10} + \rho_2) r^{n/10 + 2\rho_1 + \rho_2 - 2\alpha - 1} +$$

$$2m_0 C_0 \rho_1 \sum_{n=1}^{\infty} C_n r^{n/10 + 2\rho_1 + \rho_2 - 2\alpha - 1} - 2C_0 r^{2\rho_2 + \rho_1 - 2\alpha - 1} \sum_{n=1}^{\infty} C_n r^{n/10} \sum_{n=1}^{\infty} m_n(\frac{n}{10} + \rho_2) r^{n/10} +$$

$$m_0 \rho_1 r^{2\rho_2 + \rho_1 - 2\alpha - 1} \sum_{n=1}^{\infty} C_n^* r^{n/10} - r^{3\rho_2 - 2\alpha - 1} \sum_{n=1}^{\infty} C_n^* r^{n/10} \sum_{n=1}^{\infty} m_n(\frac{n}{10} + \rho_2) r^{n/10} = \tag{19}$$

$$A_1(m_0 g_0 r^{\rho_1 - \alpha - 1} + r^{\rho_2 - \alpha - 1} \sum_{n=1}^{\infty} m_n g_n r^{n/10}) + B_1 r^{-2}.$$

For the identity in the obtained Equation (19), equality of coefficients at the same powers «*r*» is necessary.

When $n = 0$, the equality of degrees on the left and right sides is fulfilled when

$$3\rho_1 - 2\alpha - 1 = -2$$

$$\rho_1 = \frac{2\alpha - 1}{3}$$

Since the structure of solution (12) implies $\rho_1 < 0$, therefore

$$0 < \alpha < 0.5.$$

When $\alpha = 0.2$, we will have $\rho_1 = -0.2$. When $n = 1, 2, 3\ldots$ the equality of degrees «*r*» on the left and right sides is (19) determined by a ratio

$$3\rho_2 - 2\alpha - 1 = \rho_2 - \alpha - 1,$$

from which we obtain

$$\rho_2 = \frac{\alpha}{2}.$$

Assuming $\alpha = 0.2$, then $\rho_2 = 0.1$. Thus, for $\rho$ we obtain

$$\rho = \begin{cases} \rho_1 = \frac{2\alpha - 1}{3}, & \text{where } n = 0, \\ \rho_2 = \frac{\alpha}{2}, & \text{where } n = 1, 2\ldots \end{cases}.$$

The requirement of identity in equality (19) implies the equality of the coefficients at the corresponding powers. Due to the complexity of the structure of relation (19), all subsequent stages of the proof require fixing the values of $\alpha, \rho_1, \rho_2$. Let us carry out the subsequent steps with the specified values of the parameters $\alpha = 0.2, \rho_1 = -0.2, \rho_2 = 0.1$

For the specified values of the parameters, relation (19) is reduced to a form in which the expressions of the sums contain powers of «$r$» of the same sign for the simplicity of obtaining recurrence relations for finding $m_n$

$$C_0^2 m_0 0.2 r^{-2} - C_0^2 \sum_{n=1}^{8} m_n \left(\frac{n+1}{10}\right) r^{n/10-1.7} - C_0^2 \sum_{n=9}^{\infty} m_n \left(\frac{n+1}{10}\right) r^{n/10-1.7} +$$

$$2m_0 C_0 0.2 \sum_{n=1}^{\infty} C_n r^{n/10-1.7} - 2C_0 r^{-0.4} \sum_{n=1}^{\infty} C_n r^{(n-1)/10} \sum_{n=9}^{\infty} m_n \left(\frac{n+1}{10}\right) r^{(n-9)/10} +$$

$$0.2 m_0 r^{-1.2} \sum_{n=1}^{\infty} C_n^* r^{(n-1)/10} - 2C_0 \sum_{n=1}^{\infty} C_n r^{(n-1)/10} \sum_{n=1}^{8} m_n \left(\frac{n+1}{10}\right) r^{n/10-1.3} -$$

$$\sum_{n=9}^{\infty} m_n \left(\frac{n+1}{10}\right) r^{(n-9)/10} \sum_{n=1}^{\infty} C_n^* r^{(n-1)/10} - \sum_{n=1}^{\infty} C_n^* r^{(n-1)/10} \sum_{n=1}^{8} m_n \left(\frac{n+1}{10}\right) r^{(n-9)/10} =$$

$$A_1 \left(m_0 g_0 r^{-1.4} + \sum_{n=1}^{\infty} m_n g_n r^{n/10-1.1}\right) + B_1 r^{-2}. \tag{20}$$

With the least degree $r^{-2}$ of the left and the right side of the Equation (20) we obtain

$$0.2 m_0^3 g_0^2 = B_1,$$

from which we obtain

$$m_0 = \sqrt[3]{\frac{B_1}{0.2 g_0^2}}.$$

Analysis of the left-hand side of expression (20) indicates the following degree $r^{-1.6}$. Taking into account this degree in the right-hand side of Equation (20), we obtain

$$(0.2 m_0 2 C_0 C_1 - 0.2 C_0^2 m_1 = 0.$$

Or, after transforming the expression, we have

$$0.2 m_0^2 m_1 g_0 (2g_1 - g_0) = 0.$$

From the latter, it follows that $m_1 = 0$. Similarly, for the degree $r^{-1.5}$ we obtain from (20)

$$0.2 m_0 2 C_0 C_2 - 0.3 C_0^2 m_2 = 0,$$

or

$$m_2 (0.4 m_0^2 g_0 g_2 - 0.3 m_0^2 g_0^2) = 0.$$

The last one implies $m_2 = 0$.

We continue this process for subsequent degrees «$r$» and find expressions for the coefficients $m_n$. We present non-zero expressions for the coefficients $m_n$:

$$m_3 = \frac{A_1}{2m_0 g_3^2 - 0.4}, \quad m_9 = \frac{m_3 g_3 (A_1 - 0.2 m_0 m_3 (g_3 - 8g_0))}{m_0^2 g_0 (0.4 g_9 - g_0)},$$

$$m_{15} = \frac{m_9 (A_1 g_9 - 0.4 m_3 m_0 g_9 g_3 + 0.8 m_3 m_0 g_9 g_0 + 2 m_3 m_0 g_3 g_0) + 0.4 m_3^2 g_3^2}{0.4 m_0^2 g_0 (g_{15} - 4g_0)},$$

$$m_{21} = \frac{m_{15} (A_1 g_{15} - 0.4 m_3 m_0 g_3 + 0.8 m_3 m_0 g_0 + 3.2 m_0 g_0 g_3)}{0.2 m_0^2 g_0 (g_{21} - g_0)} - \tag{21}$$

$$\frac{-m_9 g_9 (0.2 m_9 m_0 g_9 - 2 m_9 m_0 g_0 - 0.8 m_3^2 g_3) - 2 m_3^2 g_3^2}{0.2 m_0^2 g_0 (g_{21} - g_0)},$$

$$m_{27} = \frac{m_{21}g_{21}(5A_1 + 4m_3m_0g_0) + m_{15}g_{15}(8m_3^2g_3 + 10m_9m_0g_0 + 4m_{15}m_3g_3 - 2m_9m_0g_9)}{2m_0^2g_0(g_{27} - 7g_0)} +$$

$$\frac{m_9g_9(4m_3m_9g_9 + 16m_{15}m_0g_0) + 10m_9m_3g_3}{2m_0^2g_0(g_{27} - 7g_0)},$$

$$m_{33} = \frac{1}{m_0^2g_0(2g_{33} - 17g_0)}(m_{27}(5A_1g_{27} - 2m_0g_{27}m_3g_3 + 4g_{27}m_3m_0g_0 +$$

$$28m_3g_3m_0g_0) + m_{21}(10m_0m_9g_0 - 2m_0m_9g_9 + 2m_3^2g_3 + 11m_3^2g_3^2 + 22m_0g_0) +$$

$$m_{15}(4m_3m_9g_9 + 16m_0m_{15}g_0 - m_0m_{15}g_{15} + 16m_3m_9g_9) + m_9^3g_9^2).$$

Thus, we obtain unambiguous expressions of the coefficients $m_n$.

Table 1 provides information on the values of the coefficients $m_n$.

Non-zero values of the coefficients $m_n$ are indicated in Table 1 (third line). The arrow, line 3, points to the next non-zero coefficient $m_n$ in the second line of Table 1. Table 1 clearly illustrates the pattern of non-zero values.

**Table 1.** Degrees «*r*» and numbers of coefficients $m_n$.

| Degree «*r*» | −0.2 | 0.2 | 0.3 | 0.4 | 0.5 | 0.6 | 0.7 | 0.8 | 0.9 | 1.0 | 1.1 | 1.2 | 1.3 | 1.4 | 1.5 |
|---|---|---|---|---|---|---|---|---|---|---|---|---|---|---|---|
| Number $m_n$ | 0 | 1 | 2 | 3 | 4 | 5 | 6 | 7 | 8 | 9 | 10 | 11 | 12 | 13 | 14 |
| Number $m_n$ in the right side of the Equation (20) | ↑ $B_1$ | - | - | ↑ 0 | - | - | - | - | - | ↑ 3 | - | - | - | - | - |
| Degree «*r*» | 1.6 | 1.7 | 1.8 | 1.9 | 2.0 | 2.1 | 2.2 | 2.3 | 2.4 | 2.5 | 2.6 | 2.7 | 2.8 | 2.9 | 3.0 |
| Number $m_n$ | 15 | 16 | 17 | 18 | 19 | 20 | 21 | 22 | 23 | 24 | 25 | 26 | 27 | 28 | 29 |
| Number $m_n$ in the right side of the Equation (20) | ↑ 9 | - | - | - | - | - | ↑ 15 | - | - | - | - | - | ↑ 21 | - | - |
| Degree «*r*» | 3.1 | 3.2 | 3.3 | 3.4 | 3.5 | 3.6 | 3.7 | 3.8 | 3.9 | 4.0 | 4.1 | 4.2 | 4.3 | 4.4 | 4.5 |
| Number $m_n$ | 30 | 31 | 32 | 33 | 34 | 35 | 36 | 37 | 38 | 39 | 40 | 41 | 42 | 43 | 44 |
| Number $m_n$ in the right side of the Equation (20) | - | - | - | ↑ 21 | - | - | - | - | - | ↑ 33 | - | - | - | - | - |

Based on the analysis of the expressions of the coefficients $m_n$, the expression (21), an assessment hypothesis is constructed:

$$\left| m_{6(n-1)+3} \right| \leq \frac{5^n A_1 (A_1 + B_1 + 1)^{n-1} g_{6(n-2)+3}}{B_1^{2n-1} |g_3 - g_0|^{2n-1}}. \tag{22}$$

To prove the estimate in the general case, it is necessary to compile a recurrence relation for the coefficient $m_{6n+3}$ according to Equation (20). To perform this, we determine the degree «*r*» by the index of the coefficient $(6m + 3)$ and then, according to the value of this degree from Equation (20), we assemble an expression for determining $m_{6n+3}$. Thus we obtain

$$0.4m_0C_0C_{6m+3} + 0.2m_0C_{6n-1}^* - 2C_0(C_{6n+1}m_10.2 + C_{6n}m_20.3 + C_{6n-1}m_30.4 + C_{6n-2}m_40.5 +$$

$$C_{6n-3}m_50.6 + C_{6n-4}m_60.7 + C_{6n-5}m_70.8 + C_{6n-6}m_80.9) - C_0^2m_{6n+3}(0.6n + 0.4) -$$

$$2C_0C_{6n}m_9 - m_9C_{6n-4}^* - (0.9m_8C_{6n-5}^* + 0.8m_7C_{6n-4}^* + 0.7m_6C_{6n-3}^* + 0.6m_5C_{6n-2}^* +$$

$$0.5m_4C_{6n-1}^* + 0.4m_3C_{6n}^* + 0.3m_2C_{6n+1}^* + 0.2m_1C_{6n+2}^*) = A_1m_{6n-3}g_{6n-3}. \tag{23}$$

At the next stage, we discard expressions with zero values $m_n$, including even index values. Equation (23) is simplified:

$$0.4m_0C_0C_{6m+3} + 0.2m_0C^*_{6n-1} - C^2_0 m_{6n+3}(0.6n + 0.4) - m_9 C^*_{6n-4} = A_1 m_{6n-3} g_{6n-3}. \quad (24)$$

We transform (24), solve with respect to a $m_{6n+3}$ and write out $C^*_{6n-1}$, $C^*_{6n-4}$:

$$m_{6n+3}m^2_0 g_0(0.4g_{6n+3} - (0.6n + 0.4)g_0) = A_1 m_{6n-3} g_{6n-3} - 0.2m_0(2C_1 C_{6n-1} + 2C_2 C_{6n-2} +$$

$$2C_3 C_{6n-3} + 2C_4 C_{6n-4} + 2C_5 C_{6n-5} + \ldots + 2C_9 C_{6n-9} + \ldots + 2C_{6n/2} C_{(6n-1)/2}) +$$

$$m_9(2C_1 C_{6n-4} + 2C_2 C_{6n-5} + 2C_3 C_{6n-6} + \ldots + C^2_{3n-2}).$$

Or, after removing the zero values of the terms on the right side of the last equation, we obtain

$$m_{6n+3}m^2_0 g_0(0.4g_{6n+3} - (0.6n + 0.4)g_0) = A_1 m_{6n-3} g_{6n-3} - 0.2m_0(2C_3 C_{6n-3} + 2C_9 C_{6n-9} +$$

$$2C_{15} C_{6n-15} + 2C_{21} C_{6n-21} + 2C_{33} C_{6n-33} + \ldots + 2C_{6n/2} C_{(6n-1)/2}). \quad (25)$$

An extended expression of the Formula (24) is presented afterwards. Taking into consideration the rules established in Table 1 as well as the number of zero and non-zero coefficients, the number of terms in the right side of the Equation (25) in the expression under round brackets will contain no more than $(n-1)/2$ terms. Taking into consideration that each term will contain the expression $(A_1 + B_1 + 1)$ is raised to the power $(n-2)$.

$$|g_3 - g_0| \leq |0.4g_{6n+3} - (0.6n + 0.4)g_0|.$$

This estimate can be verified using Formula (15) for calculating $g_n$, given above. So from (25), taking into account (22) and the value $m_0$ we will have

$$|m_{6n+3}| \leq \frac{5^{n+1}A_1(A_1 + B_1 + 1)^n g_{6(n-1)+3}}{B_1^{2n+1}|g_3 - g_0|^{2n+1}} = V_{6n+3}. \quad (26)$$

Let us consider the series

$$r^{\rho_2} \sum_{n=1}^{\infty} V_n r^{n/10}, \quad (27)$$

which, by virtue of (26), is a majorant for the regular part of the series (12).

Based on the sufficient criterion for the convergence of power series, we obtain the region of convergence for the series (27)

$$r < \left( \frac{(g_3 - g_0)^2 B_1^2}{5(A_1 + B_1 + 1)} \right)^{5/3} = \rho_3 \quad . \quad (28)$$

Therefore, the correct part of the series (12) also converges in this region.

Thus, we complete the proof of Theorem 1, the existence and uniqueness of the solution in the neighborhood of a fixed singular point.

**Theorem 1.** *For the singular problem* (10), (11) *there is a unique solution in the form* (12) *in the domain*

$$r < \left( \frac{(g_3 - g_0)^2 B_1^2}{5(A_1 + B_1 + 1)} \right)^{5/3} = \rho_3$$

*when* $A_1 < B_1$, $B_1 > 1$, $0 < \alpha < 0.5$,

$$\rho = \begin{cases} \rho_1 = \frac{2\alpha - 1}{3}, & \text{where } n = 0, \\ \rho_2 = \frac{\alpha}{2}, & \text{where } n = 1, 2 \ldots \end{cases}$$

The proven theorem allows us to construct an analytical approximate solution to problem (10) and (11) in the form

$$P_N(r) = m_0 r^{\rho_1} + r^{\rho_2} \sum_{n=1}^{N} m_n r^{n/10}. \tag{29}$$

and obtain its a priori error estimate.

**Theorem 2.** *When the conditions of Theorem 1 are met for the analytical approximate solution (29) of problem (10) and (11) and the selected parameters $\alpha = 0.2$ and $\rho_2 = 0.1$, the error estimate is valid*

$$\Delta P_N(r) \leq \frac{5^{N+1} A_1 (A_1 + B_1 + 1)^N g_{6N-3} r^{0.6N+0.4}}{B_1^{2N} (g_3 - g_0)^{2N+1}} \left( \frac{1}{(1 - (5(A_1 + B_1 + 1) r^{0.6} / (B_1^2 (g_3 - g_0)^2)))} \right)$$

*in the region $r < \rho_3$, where*

$$\rho_3 = \left( \frac{(g_3 - g_0)^2 B_1^2}{5(A_1 + B_1 + 1)} \right)^{5/3}.$$

**Proof.** The estimates for $m_n$ in Theorem 1 allow us to obtain an a priori estimate of the analyticity of the approximate solution (29)

$$\Delta P_n(r) = |P(r) - P_N(r)| = \left| \sum_{n=N+1}^{\infty} m_{6(n-1)+3} r^{0.6n-0.2} \right| \leq \left| \sum_{n=N+1}^{\infty} V_{6(n-1)+3} r^{0.6n-0.2} \right| =$$

$$\left| \frac{5^{N+1} A_1 (A_1 + B_1 + 1)^N g_{6(N-1)+3} r^{0.6N+0.4}}{B_1^{2N+1} (g_3 - g_0)^{2N+1}} + \frac{5^{N+2} A_1 (A_1 + B_1 + 1)^{N+1} g_{6(N-1)+3} r^{0.6N+1}}{B_1^{2N+3} (g_3 - g_0)^{2N+3}} + \ldots + \right.$$

$$\left. \frac{5^{N+k} A_1 (A_1 + B_1 + 1)^{N+k-1} g_{6(N+k)-3} r^{0.6(N+k)-0.2}}{B_1^{2(N+k)-1} (g_3 - g_0)^{2(N+k)-1}} + \ldots \right| \leq$$

$$\frac{5^{N+1} A_1 (A_1 + B_1 + 1)^N g_{6N-3} r^{0.6N+0.4}}{B_1^{2N} (g_3 - g_0)^{2N+1}} \left( \frac{1}{B_1} + \frac{5(A_1 + B_1 + 1) g_{6N+3} r^{0.6}}{B_1^3 (g_3 - g_0)^2 g_{6N-3}} + \right.$$

$$\left. \frac{5^2 (A_1 + B_1 + 1)^2 g_{6N+9} r^{1.2}}{B_1^5 (g_3 - g_0)^4 g_{6N-3}} + \ldots + \frac{5^{k-1} (A_1 + B_1 + 1)^{k-1} g_{6(N+k)-3} r^{0.6k-0.6}}{B_1^{2k+1} (g_3 - g_0)^{2k} g_{6N-3}} + \ldots \right). \tag{30}$$

Taking into account the condition

$$\frac{B_1 g_{6N-3}}{g_{6(N+k)-3}} > 1$$

for $k = 2, 3 \ldots$, from (30) we obtain

$$\Delta P_N(r) \leq \frac{5^{N+1} A_1 (A_1 + B_1 + 1)^N g_{6N-3} r^{0.6N+0.4}}{B_1^{2N} (g_3 - g_0)^{2N+1}} \left( 1 + \frac{5(A_1 + B_1 + 1) r^{0.6}}{B_1^2 (g_3 - g_0)^2} + \right.$$

$$\left. \frac{5^2 (A_1 + B_1 + 1)^2 r^{1.2}}{B_1^4 (g_3 - g_0)^4} + \ldots + \frac{5^k (A_1 + B_1 + 1)^k r^{0.6k}}{B_1^{2k} (g_3 - g_0)^{2k}} + \ldots \right) \leq$$

$$\frac{5^{N+1} A_1 (A_1 + B_1 + 1)^N g_{6N-3} r^{0.6N+0.4}}{B_1^{2N} (g_3 - g_0)^{2N+1}} \left( \frac{1}{(1 - (5(A_1 + B_1 + 1) r^{0.6} / (B_1^2 (g_3 - g_0)^2)))} \right)$$

in the region

$$r < \left( \frac{B_1^2 (g_3 - g_0)^2}{5(A_1 + B_1 + 1)} \right)^{5/3} = \rho_3.$$

Thus, we complete the proof of Theorem 2.   □

### 3. Results Discussion

Let us consider problem (10) and (11), where $A_1 = 0.0000003$, $B_1 = 737.12$.

According to Formula (28), we determine $\rho_3$, $\rho_3 = 0.76618$, $r_1 = 0.4$ falls within the domain for Theorem 2.

The analytical approximate solution (29) takes the form as follows

$$P_{27}(r) = 16.691428474r^{-0.2} + 5.90354928 \times 10^{-7}r^{0.4} - 6.20530236 \times 10^{-14}r^1 +$$

$$6.6 \times 10^{-21}r^{1.6} - 7.28 \times 10^{-28}r^{2.2} + 1.64 \times 10^{-28}r^{2.8}. \tag{31}$$

The calculations are given in Table 2.

**Table 2.** Numerical experiment calculations.

| $r_1$, m. | $P_{27}(r_1)$, Pa. | $\Delta P_{27}(r_1)$ | $\Delta$ |
|---|---|---|---|
| 0.4 | 20.0484829874667 | $1.231285 \times 10^{-11}$ | $1.1138 \times 10^{-12}$ |

In this case, $P_{27}(r_1)$ is an analytical approximate solution (29); $\Delta P_{27}(r_1)$ is an a priori estimate of the error $P_{27}(r_1)$; $\Delta$—is an a posteriori estimate of the error $P_{27}(r_1)$.

For the $\Delta$ an analytical approximate solution specified in the structure (29) is required $N = 33$. The sum from the 27th to the 33rd terms in the structure (29) does not exceed the value $1.1138 \times 10^{-12}$. Therefore, the a priori error estimate $P_{27}(r_1)$ does not exceed the value $\Delta = 1.1138 \times 10^{-12}$.

A graphical interpretation of the analytical approximate solution (29) in the vicinity of a fixed singular point is given in Figure 1.

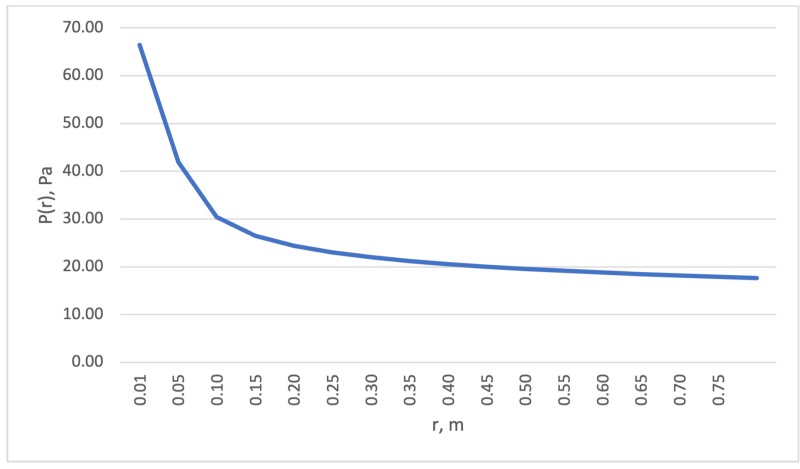

**Figure 1.** Graphical interpretation of the analytical approximate solution (31).

### 4. Conclusions

The paper presents a proof of the existence and uniqueness theorem for a solution to the singular Cauchy problem (10) and (11) for a nonlinear fractional differential equation in the neighborhood of a fixed singular point. An analytical approximate solution is constructed, and an a priori error estimate is obtained. A formula is obtained for calculating the region in which the proven theorem 1 works. A technology for optimizing an a priori error using an a posteriori error is demonstrated. The results presented in the paper are the first stage of the study of nonlinear differential equations with fractional derivative. The next stage involves the study of an analytical approximate solution in the region of analyticity of the solution, and then in the vicinity of a moving singular point.

**Funding:** This research received no external funding.

**Data Availability Statement:** The statistical data presented in the article does not require copyright. They are freely available and are listed at the reference address in the bibliography.

**Acknowledgments:** The author expresses gratitude to the reviewers for valuable comments that helped improve the content of the article, as  well as to the editors of the journal for their positive attitude towards the work.

**Conflicts of Interest:** The author declare no conflicts of interests.

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
