# Peer review of "Singular Cauchy Problem for a Nonlinear Fractional Differential Equation"

_mathematics, doi:10.3390/math12223629_

Round 1

Reviewer 1 Report

Comments and Suggestions for Authors

The paper presents proof of the existence and uniqueness of the theorem for a solution to the singular Cauchy problem for a nonlinear fractional differential equation in the neighborhood of a fixed singular point. An analytical approximate solution is proposed. My observations for this work re as follows:

1. The transformation made in Eq. (9) is unclear. Please elaborate to proceed for the next mathematical expressions after Eq. (9).

2. Elaborate on the term B_1 in Table 1.

3. The expression after Eq. (25) needs more elaboration.

4. The proof of Theorem 1 is missing. Please add.

5. Some immediate future work should be mentioned in the conclusion section.

Author Response

Responses to Reviewer's Comments 1

I thank the reviewer for the professional review of the article text, which allowed me to improve the presented material for understanding the research results.

         Question 1.

The transformation made in Eq. (9) is unclear. Please elaborate to proceed for the next mathematical expressions after Eq. (9).

Author's response: The typo in formula (7) has been corrected. Formula (9) is obtained from formula (7).

Question 2.

Elaborate on the term B_1 in Table 1.

Author's response: B1 appears from formula (21), from the condition of equality of coefficients at the power r-2.

Question 3.

The expression after Eq. (25) needs more elaboration.

Author's response: The remark has been taken into account and an explanation has been given in the text.

Question 4.

The proof of Theorem 1 is missing. Please add.

Author's response: The proof of the theorem is presented in a non-standard way before the formulation of Theorem 1 and begins after the formulation of the Cauchy problem (10)-(11).

Question 5.

Some immediate future work should be mentioned in the conclusion section.

Ответ автора: Замечание было принято во внимание. В заключении сформулированы следующие задачи для решения.

Reviewer 2 Report

Comments and Suggestions for Authors

Comments on the Quality of English Language
    • Changed "including" to "involving" for precision.

      Changed "theorem of existence of uniqueness of a solution" to "theorem of existence and uniqueness of a solution."

      Replaced "built" with "constructed" for clarity.

      Replaced "area where the proven theorem works" with "region where the proven theorem is applicable."

      Replaced "both digital and graphical versions" with "both digital and graphical forms."

      "A" before "singular Cauchy problem is considered."

      Used "the" consistently before terms such as "theorem," "a priori estimate," and "theoretical results."

Author Response

Responses to Reviewer's Comments 2

I thank the reviewer for the professional review of the article text, which allowed me to improve the presented material for understanding the research results.

Question 1.

The original contributions should be presented more clearly in the final paragraph of the INTRODUCTION section. Any improvements or new findings should be thoroughly described in this part, as the text does not currently address the advantages of the work.

Author's response: The remark has been taken into account in the revised version of the article. The solved problems and the novelty of the study for the equation with a fractional derivative in this version are indicated.

Question 2.

A brief outline of the paper’s structure could be added at the end of the Introduction.

Author's response: The comment has been taken into account in the revised version.

Question 3.

Does the author clearly explain the meaning and significance of the variables α, ρ1, and ρ2? Are these terms adequately defined for readers?

Author's response: The meanings of the parameters are explained throughout the text.

Question 4.

Are the derivations and steps in equation (19) and the following calculations mathematically rigorous and logically consistent? Do any steps require further clarification?

Author's response: All calculations after equation (19) are logically consistent and mathematically justified. This technology is performed for fractional equations for the first time, and for ordinary nonlinear equations it has a fairly wide application, confirmed by a sufficient number of author publications in articles indexed in WoS and Scopus.

Question 5.

The text mentions obtaining recurrence relations for finding mn. Are these recurrence relations well-justified, and are they used effectively in the derivation?

Author's response: The obtained recurrence relations are directly related to the proof of the theorem.

Question 6.

Is the terminology consistent throughout the derivations? For instance, are terms like ”equality of degrees” and ”identity of coefficients” used precisely and consistently?

Author's response: All terminology is used as intended during the proof of theorems.

Question 7.

Given the complexity of the structure of relation (19), is the explanation of fixing the parameters α, ρ1, and ρ2 sufficiently detailed? Could additional clarification benefit readers?

Author's response: Fixing the parameters in equation (19) is necessary to obtain the correct recurrence relation.

Question 8.

Are the results obtained for ρ1 and ρ2 in line with the assumptions made in the analysis? Are there any potential limitations or alternative approaches the author should address?

Author's response:  The values ​​of the parameters depend on the order of the fractional derivative, this is observed throughout the text of the proof.

Question 9.

The authors are encouraged to report CPU running times in the Numerical Results section.

Author's response: The revised version notes what was used when carrying out calculations.

Question 10.

The authors should specify which software was used to produce the results presented in the paper.

Author's response: The revised version notes what was used when carrying out calculations.

Question 11.

The list of literature should be expanded by including classical works on the application: https://doi.org/10.1007/s40314-022-01934-y http://dx.doi.org/10.22436/jmcs.028.02.03 http://dx.doi.org/10.22436/jmcs.031.04.02

Author's response:  The article deals with an analytical approximate solution in the vicinity of a fixed singular point and there is no connection with numerical methods in this case.

Question 12.

It would be better to write the conclusion section in the past tense rather than the present tense.

Author's response: The comment has been taken into account in the revised version of the article.
